# Circadian Intervention Improves Parkinson’s Disease and May Slow Disease Progression: A Ten Year Retrospective Study

**DOI:** 10.3390/brainsci14121218

**Published:** 2024-11-30

**Authors:** Gregory Willis, Takuyuki Endo, Murray Waldman

**Affiliations:** 1The Bronowski Clinic, The Bronowski Institute of Behavioral Neuroscience, Woodend, VIC 3442, Australia; 2Department of Neurology, Osaka Toneyama Medical Center, 5-1-1, Toneyama, Toyonaka 560-8552, Osaka, Japan; 3Sunnex Biotechnologies, 657-167 Lombard Ave, Winnipeg, MB R3B 0V3, Canada

**Keywords:** circadian, Parkinson’s disease, sleep, fatigue, motor function, depression, prodromal, bright light therapy, progressive degeneration

## Abstract

Background: The involvement of the circadian system in the etiology and treatment of Parkinson’s disease (PD) is becoming an increasingly important topic. The prodromal symptoms of PD include insomnia, fatigue, depression and sleep disturbance which herald the onset of the primary symptoms of bradykinesia, tremor and rigidity while robbing patients of their quality of life. Light treatment (LT) has been implemented for modifying circadian function in PD but few studies have examined its use in a protracted term that characterizes PD itself. Methods: The present exploratory study monitors the effect of LT over a 10 year course of PD in the context of ongoing circadian function. Results: Improvement in circadian based symptoms were seen soon after LT commenced and continued for the duration of the study. Improvement in motor function was more subtle and was not distinguishable until 1.2 years after commencing treatment. Improvement in most motor and prodromal symptoms remained in steady state for the duration of the study as long as patients were compliant with daily use. Conclusions: The sequence of improvement in prodromal symptoms and motor function seen here parallels the slow, incremental repair process mimicking the protracted degenerative sequelae of PD that extends over decades. This process also emulates the slow incremental improvement characterizing the reparative course seen with circadian symptoms in other disorders that improve with LT. Recent findings from epidemiological work suggest that early disruption of circadian rhythmicity is associated with increased risk of PD and the present findings are consistent with that hypothesis. It is concluded that intervening in circadian function with LT presents a minimally invasive method that is compatible with internal timing that slows the degenerative process of PD.

## 1. Introduction

For more than four decades, simulating, stimulating, protecting, repairing or replacing damaged neurons in the nigro-striatal dopamine (NSD) system have been at the primary focus of scientific research into the cause and treatment of Parkinson’s disease (PD). However, it has become clear that this approach has limitations, as declining efficacy, polypharmacy and adverse effects of dopamine (DA) replacement rob patients of their quality of life, rendering treatment problematic. For these reasons the search continues for less invasive approaches to achieve more effective management of the disease [1].

Recent work demonstrating circadian involvement in PD [2] has been encouraging as it provides new strategies for understanding disease etiology and presents innovative treatment options. While many novel circadian interventions remain highly speculative [3], strategic light exposure is minimally invasive and is most promising [4,5,6,7,8]. This is based on the proposition that light is the most effective “zeitgeber” or timekeeper for realigning circadian phase. To date there have been twelve studies undertaken using various light source types and light presentation regimens to examine the therapeutic feasibility of this approach.

In the original open label trial implementing bright light therapy in PD, light was employed to reduce the adverse side effects of DA replacement and for treating prodromal depression [4]. Since that time there have been ten further trials consisting of two open label [6,9,10], two retrospective [7,11], and six randomized, blinded control studies [5,8,12,13,14,15] implementing a variety of sleep monitoring assessments and procedures to assess sleep as an index of circadian involvement [11,12,13,15]. The reported benefits of light treatment (LT) in repairing prodromal PD vary considerably.

Limitations in interpretation of these studies due to variations in the light sources employed, time and regularity of light administration, and intensity and duration of treatment make comparisons across studies difficult. For this reason, further definitive research will provide a firm foundation for defining the effective application of light to achieve the optimal therapeutic response. This is essential in a complex, chronic disease such as PD requiring extended treatment over several decades. On this basis, we undertook a ten year study examining the long-term effects of light treatment (LT) on the circadian features of PD by examining an array of motor, sleep and other prodromal symptoms. While implementation of a controlled trial is problematic for a study of ten years duration, some parameters studied here were blinded by the very nature of the procedure employed, while verification of each patient’s testimony by their carer, spouse or partner during the structured clinical interview helped to objectify data collection. In addition, the method of long-term assessment of prodromal and primary symptoms used in the present study has been verified previously [7] and confirmed in a previous controlled trial [8]. The present study provides a detailed, long-term exploration of the effects of ongoing LT in PD in the context of circadian function that is routinely studied in the short term.

## 2. Materials and Methods

### 2.1. Participants

Histories from one hundred and nineteen patients diagnosed previously with idiopathic PD that had attended or were currently attending the Bronowski Clinic were included in the study. The Bronowski Ethical Standards Committee approved a waiver of consent as stipulated under the guidelines of The National Health and Medical Research Council of Australia (NH and MRC) in the National Statement on Ethical Conduct in Human Research 2023. Patient records were de-identified via the ethical handling of histories and data storage as approved and monitored on a regular basis throughout the study. Trial Registration: This trial was an exploratory, retrospective trial that was registered with the Australian and New Zealand Clinical Trial Registry. Registration number ACTRN12623000147684.

The only eligibility criterion for inclusion in the study was a previous diagnosis of idiopathic PD by a qualified neurologist or Movement Disorder Specialist. Given that the purpose of the Bronowski Clinic is first to serve all PD patients and that the data collection was observational, there were no inclusion criteria. Patients that were currently receiving LT at the Bronowski Clinic since 2012 or later were included. No exclusion criteria were exercised in that, while exceptional comorbidities were noted, the object of the exploratory nature of the research was to determine what relief LT might provide through difficult, complex diagnoses and complex treatment regimens. The demographics of all patients included in the study are described in Table 1.

### 2.2. Treatment Description

In most cases, light was administered by utilizing a light source containing fluorescent tubes (Apollo BL-6; Medi-light; or equivalent, without ultraviolet emission) with an emission spectra similar to that described in Willis et al., 2018 [8]. Since melatonin is secreted primarily at night with these patients being described as phase advanced [10,17,18,19], exposure was prescribed for 1 h between the hours of 18:00 and 22:00 h., usually occurred just prior to bedtime. As a general rule of thumb, light was administered at a dose ranging from 3000 to 4000 LUX achieved by placing the device 0.8 to 1.0 m from the bridge of the nose to the front of the diffuser at an angle ≤30° to the mid-sagittal plane of the head. Light was systematically offered to all idiopathic PD patients and implemented with an exploratory trial session.

### 2.3. Randomization and Blinding

When ongoing assessments and archived histories were made accessible, each patient’s history was coded and then pooled. While the trial was, for the most part, regarded as a historical, longitudinal, open label study, various parts were blinded to the experimenter and to the patients during the course of the study. In the instances where blinding occurred, the condition of each patient was assessed and compared to their score obtained on the previous evaluation. The patient’s condition was verified, in almost all cases, by the carer, partner or spouse describing the condition of the patient over time, to avoid the patient’s perception being tainted by acute events just prior to the evaluation. This served to provide a more objective opinion of the patient’s progressive state by someone who experienced the patient on a regular basis. This helped to counter the anosognosia, or lack of self-awareness, that these patients commonly exhibit. Furthermore, at no time during or after each evaluation did the patient receive any feedback as to the numerical value assigned on the Likert Scale describing the severity of symptoms that the patients displayed. It is acknowledged that, while this may not have completely overcome the potential bias that can influence the value assigned by the clinician at any given point in the evaluation, the input of the attending carer helped to minimize clinician bias.

Implementation of the timed motor tests (TMTs) was a more objective means of measuring motor function in the upper and lower limbs of these patients in that it does not require a clinical judgement. These tests included the elbow-to-fist (ETF) arm test and floor to knee (FTK) leg test that have been described previously [6,7,8]. The validity of the TMTs has also been confirmed in a recent randomized, double blind, placebo controlled trial examining the efficacy of LT in PD [8]. In the present study, patients were not permitted access to any of their performance times during the course of their treatment or after they left the program. Patients remained in the program for a time varying from 3 months up to more than 10 years and were not permitted access to any other patient’s history during that time. In this regard patients were blinded as to their own performance at each session and to the performance of other patients in the trial. Blinding the patients to values assigned to all numerical parameters occurred for all clinical parameters measured during the course of the study. In consideration of the weakness of the Unified Parkinson’s disease Rating Scale (UPDRS) described recently [20,21], our implementation of TMTs and Likert Scales on various parameters provided a more valid and reliable account of disease progression, such as that occurring with the TMTs, and this can be easily applied in a clinical setting over long durations. 

### 2.4. Study Design

Patients were assessed on all parameters on their first scheduled visit to the clinic. These measures were used as baseline readings with all subsequent readings compared to these values. Assessments were scheduled at intervals of 1, 2, 3, 5 and 6 months sequentially and then every 4–6 months thereafter, as described previously [7], for as long as each patient remained in the program over a ten year period. A Likert Scale was used to score patients on a ten point scale as being slight to severe and this has been used extensively in previous studies [6,7,8,22]. All clinical evaluations and tests were performed between the hours of 0900 and 1700 h. All patients continued with their routine use of LT at the end of the study.

### 2.5. Circadian, Sleep and Related Parameters

To examine sleep patterns and the presence of sleep disturbance three measures were employed to achieve cross-confirmation of sleep architecture [23] and of the quantity of sleep achieved using a method described previously [11]. These included (i) recording sleep in a sleep diary completed each morning prior to starting treatment and for at least 40 days into the program, (ii) providing a spontaneous report on sleep parameters during a structured interview [8] and (iii) completing a circadian record at the conclusion of each formal assessment [8,11]. Score on the Morningness–Eveningness questionnaire (MEQ–SA) were acquired for all patients still enrolled at the time the study was completed (*n* = 26) to determine their circadian type [16]. The clinical assessment model employed encouraged a spouse, partner, carer or family member to be present during each clinical assessment to provide supplemental feedback to verify the patient’s responses for each measure.

### 2.6. RSBD

To assess and classify the severity of RSBD, the following criteria were implemented during each assessment at the regularly scheduled intervals described earlier with the diagnostic criteria defined by the International Classification of Sleep Disorders [24]. The frequency and severity of RSBD were assessed on a global basis [11,25] by the clinician during each interview and these were verified, when possible, by a spouse, partner, carer or relative as described previously for insomnia.

### 2.7. Statistical Design and Analysis

The rationale for developing the assessment scale used in the present study arose from the need to provide an easily executed and readily accessible record of these parameters for the clinician over an extended period. Limited consultation time in general and neurological practice does not readily permit the clinician to record data from lengthy assessments, and to interpret those results, over a very long-term, such as that examined in the present study. Secondly, given the very nature of the Bronowski Clinic as a research clinic, collecting data from longitudinal studies over several years required a system for synopsizing large volumes of information. Condensation of essential data into such a rating facilitated the process that was employed. Third, a condensed method for an efficient and effective data collection system from such large case series study was key in providing direction and a solid rationale for undertaking future formal controlled clinical trials to extend previous findings [7,11].

The standard statistical approach in a previous retrospective study over 5 years compared the pre- and post-treatment performance at strategic times after treatment commenced [7]. However, in this protracted study where the performance of each patient was followed for 10 years, there was a rate of patient drop out whereby the number of patients markedly diminished over the period of observation between each assessment. This was due to events such as transfer, discharge or death during the course of the study. Under professional statistical advice, non-parametric testing was implemented using the Related Measures Wilcoxon Signed Rank Test (IBM-SPSS^®^-2022) to compare pre- and post- treatment performance at time points selected a priori. To reduce the incidence of Type I errors associated with multiple testing a Bonferroni adjustment was applied whereby α levels of 0.05 (0.05/7 = 0.0071) and 0.001 (0.001/7= 0.0014) were calculated and applied for each test. The working hypothesis was generated a priori and tested at representative intervals evenly across the 10 year duration. Standard error of the mean was calculated for placement on all graphs as it permitted a meaningful visual inspection of variability across samples of a population at all-time points, including those not formally analyzed in statistical testing.

### 2.8. Determination of Responsive Versus Non-Responsive Patients

All patients that entered the clinic were included in the exploratory trial without regard for non-compliance, protocol deviations, withdrawal or anything that happened since they had been enrolled, in keeping with an intention-to-treat concept [26]. Due to the inherent lack of homogenous expression of all Parkinsonian features across the broad spectrum of symptoms characterizing this disorder, each patient was classified as either Responsive or Non-responsive to LT for each of the variables studied. The classification of each patient was achieved by finding the mean score for each variable, across all treatment sessions attended. The mean score achieved was then compared to the pre-treatment score recorded prior to commencing LT. If the mean treatment score achieved improved compared to the pre-treatment score, then the patient was classified as “responsive”. If the score was worse than the pre-treatment score, then the patient was classified as “non-responsive”. The total number of responsive and non-responsive patients were grouped for analysis for each variable at each session and at each time point, for the duration of their involvement in the study. To determine the generalizability of the effect, the number of responders vs non-responders for each variable studied was also expressed on the graphic representation.

The time points chosen for statistical analysis were selected on the basis that they depicted an array of times across the 10 year span of observation, permitting detection of short and long-term effects of LT for each of the parameters. For Figure 1, the time periods chosen for comparison with the pre score were 1 month, 0.5, 1 and 2 years and for 3–10 years. In the extended longer term data, entries were binned at 25–30 months, 30–36 months, 37–42 months and 43 to 100 months. The last four time groups were binned to enable meaningful statistical analysis as patient numbers dwindled toward the end of the study due to the aforementioned rate of natural attrition in clinic numbers. For Figures 3–7, statistical comparisons were made between pre and 1 month, 1, 2, 3, 5 and 6 to 10, 7–10 or 8–10 years. The choice of times was arbitrary as they were applied equally across responders and non-responders.

While patient compliance with the schedule assessment time did not always conform to the strategic plan as described earlier in the study design section, data were analyzed on the basis of pre-treatment values (taken as the baseline value) versus the first session on month 1 and for all representative times extending over the course of 10 years of observation as each patient remained in the program. One of the functions of the trial was to observe the duration of time that patients voluntarily remained in the study. In the present “real world” study we wanted to examine the natural rate of attrition as the number of patients dwindled over the decade examined. If the number of remaining patients naturally dwindled to less than 10, then consecutive sessions were binned to achieve a minimal number of subjects that would permit meaningful statistical analysis (i.e., *n* = 10-minimum) for each statistical comparison.

## 3. Results

### 3.1. Time to Sleep and Awaken Before and After LT

Table 2 illustrates the effect of LT on time to sleep and time to awaken in PD patients undergoing LT for up to 10 years. In the short term, when patients recorded their time to sleep and to awake in a sleep diary, 43% of patients gradually went to sleep earlier while 57% went to sleep later. A similar scenario transpired in regard to time to awaken with the same percentage showing a tendency to advance and delay their phase in the same proportions. In the longer term, a similar scenario was seen with the exception that fewer patients tended to advance their phase by awakening earlier, meaning that more of these patients (70%) in the term of up to ten years experienced a delayed phase and woke later.

### 3.2. The Amount of Sleep, Number of Awakenings and Tendancy to Fall Back After LT

Figure 1A illustrates the change in total sleep time for all patients showing a significant increase in sleep for the duration of the study. Note that, out of 114 patients, 82 patients were rated as responsive to evening LT (74%) while 26 were non-responsive. Note that the number of hours of sleep per session gradually increases for the duration of the program over a 10 year period and the pre versus post LT change was significant for up to 1.5 years and stabilized in the longer term from 3 to 10 years. Figure 1B demonstrates that the total amount of sleep achieved in the non-responsive group (*n* = 26) did not change significantly from that achieved prior to LT. At the end of the 10 year period, the total amount of sleep worsened significantly.

Figure 1C depicts the mean number of awakenings during the dark cycle for a period of up to 10 years of LT for responsive and non-responsive patients. Out of 114 patients, 65 comprising the responsive group, showed a significant decrease in the number of awakenings, suggesting less sleep fragmentation in this group. During the study period, all but one critical time point measured during the 10 year period were highly significant. The remaining 43 non-responding patients failed to improve (Figure 1D) and displayed a gradual increase in the number of awakenings compared to before LT, with the number of awakenings increasing significantly at year 1 and year 10 after entering the program.

Figure 1E shows that patients improved on falling back to sleep when they awakened during the night after LT (*n* = 56) compared to the non-responders, where performance on this parameter deteriorated by the 10th year (*n* = 52; Figure 1F). The difference between the pre-program entry and the improvement over the 10 year period was significant in the responsive group at all time periods, while there was no significant change in those that were classified as non-responsive, except at the end of the ten year period when the tendency to fall back to sleep worsened. This contributed to improved quality of sleep in the responsive patients.

The frequency plot in Figure 2 illustrates the relationship between the severity of insomnia and the tendency for LT to increase the amount of sleep during the night. Note that all patients in the study that had severe insomnia, defined as those that sleep less than 5 h per night, improved their sleep in the present of LT. This constituted 69% of the total sample of patients tested in the study. Only those patients that achieved 5.5 to 10 hrs. of sleep prior to commencing the program were non-responsive.

### 3.3. Fatigue

Figure 3 shows that that evening LT reduced the daytime fatigue in 85 of the patients in the study. The improvement was seen as early as 1 month after treatment commenced and remained significantly reduced at 6 of the 7 critical time points examined over the 10 year period. In the 26 remaining patients classified as non-responsive, there was no significant change in fatigue at any time during the 10 day duration of the study. However, it should be noted that these patients dropped out of the study earlier than the responsive patients.

### 3.4. Timed Motor Tests

Figure 4A depicts the latency to perform the ETF test in 84 of the patients that were defined as the responsive group. There was increasing improvement in the latency to perform this task for a period of up to 10 years with all seven critical time points being significantly quicker than the average pre-treatment score. Conversely, the 28 patients classified as non-responsive gradually worsened. With three of the five time points tested showing significant slowing of the response, the remaining two critical time points of comparison in this group showed no change in performance from the pretreatment score. This was partially attributed to lack of compliance and is consistent with quitting the program early in the course of treatment.

A similar response was seen for the latency to perform the FTK test, as seen in Figure 4B. A significant incremental improvement in time was required to perform this test in the responsive group (*n* = 75) at 1 month, year 1 and 3 and at years 8–10 after commencing LT. By comparison, the non-responsive group (*n* = 33; Figure 4C) showed worsening on this parameter at 1 month and at 1 and 2 years after commencing treatment, while there was no change in the other time points when compared to pre-treatment performance. Note the feint parameters [7].

### 3.5. Depression

As shown in Figure 5, patients that were diagnosed with depression as a comorbidity exhibited slight to moderate depressive symptoms when they entered the program. There was a rapid reduction in depression for this responsive group (*n* = 83) as early as the first session at one month after commencing LT. This anti-depressive trend continued for the duration of the study and was significant at all critical time points during the course of the 10 years. The non-responsive patients (*n* = 27) showed no significant change in depression during the entire 10 year course when compared to their score prior to commencing LT. Note that patients experiencing the most severe depression exhibited the most robust improvement after LT, as was observed with sleep (Figure 2) and as seen in prior controlled trials with LT [5].

### 3.6. Primary Symptoms

Figure 6A illustrates the effect of evening phototherapy on the population of PD patients over the 10 year period of observation. Note that while a significant improvement in bradykinesia was apparent for the first 3 years for all patients in the responsive group (*n* = 57), it incrementally deteriorated from the fourth to the tenth year. Overall the effect of phototherapy on bradykinesia in the non-responsive group was not significantly different when compared to pre-treatment score (*n* = 50) and there was no significant change at any critical time of measurement during the 10 year period.

Figure 6B depicts the effects of 1 h of LT on muscular rigidity in the responsive patients (*n* = 54) in the program. There was an incremental, statistically significant improvement in this parameter up to 4 years with a tendency to deteriorate slightly from the 4th to the 6th year. After this a significant improvement was seen in the 7–10th year. For the non-responsive group (*n* = 51), there was no change in four of the critical time periods for rigidity; however, at years 1 and 7–10, there was a highly significant worsening in rigidity in this subset of patients. As we observed with bradykinesia and sleep, the patients most severely affected by rigidity were those that had the most pronounced therapeutic response to evening LT exposure. Figure 6C shows the effect of phototherapy on tremor in PD patients treated for up to 10 years. The responsive group was large (*n* = 73), showing a statistically significant reduction in tremor at all critical time points of comparison during the 10 year course of treatment. Conversely, the non-responsive group (*n* = 19) showed significant worsening of tremor at 1 and 2 years after commencing LT, with the remaining times tested being unchanged from pre-treatment levels.

### 3.7. Dyskinesia

Figure 7 illustrates the effect that LT had on dyskinesia during the 10 year course of treatment in those patients entering the clinic with and without dyskinesia.

The demographics of dyskinesia for patients in the study are shown Table 3. Note that those entering the clinic with dyskinesia remained stable and showed no significant change during the 10 year course. However, those commencing the trial with no dyskinesia showed an incremental increase in dyskinesia that increased significantly for the first 3 years, then again at years 7 through 10. It is also noteworthy that the variability of dyskinesia for both groups during the fourth through tenth year of observation increased remarkably. The trend analysis shows dyskinesia decreasing over time for those attending the clinic with pre-existing dyskinesia, while those entering LT treatment without it incrementally deteriorated as the disease advanced.

### 3.8. Chronotype as It Related to Sleep and Motor Function

Table 4, Table 5, Table 6, Table 7 and Table 8 illustrate the performance of a random sample of 26 patients that were scored on the MEQ self-assessment in relation to their performance on 10 variables measured in the study. Their scores on the MEQ-SA permitted their classification into chronotypes ranging from definite morning to intermediate. All patients included in the MSQ-SA subsample were classified as intermediate to definite morning types with none being defined as evening chronotype.

Table 4 shows the change in time to fall asleep or to awaken changes in the presence of LT during the first 40 days (left columns) and over 10 years of LT (right columns).

Note that during the first 40 days the moderate and definite morning chronotypes go to bed earlier while more patients with an intermediate score are delayed or go to sleep later. In regard to awakening, most of these patients, regardless of chronotype, are more likely to delay their phase when awakening. During the 10 year period of observation (right columns), there was little difference between the numbers of patients showing an advancement or delay of phase in respect to time to fall asleep. However, LT exposure more frequently causes a delay in the sleep phase, as evidenced at the time of awakening. This suggests that the process of shifting phase occurs gradually over several years with the tendency for LT to cause a phase shift not consistent with chronotype. Table 5 depicts improvement in sleep, reduction in the number of awakenings and improvement in the tendency to fall back to sleep in a majority of patients treated with evening LT being independent of the MEQ-SA score. While all but four of the patients showed an improvement in the amount of sleep achieved after LT (left columns), the number of patients showing a reduction in the number of awakenings was slightly greater than those showing deterioration (middle columns). Improvement in the ability to fall back to sleep was affected in about 50% of the patients, while the remaining half deteriorated on this parameter (right columns).

Table 6 shows the relationship of MEQ-SA score to the effect of LT on fatigue (left columns), depression (middle columns) and dyskinesia (right columns) in our sample of PD patients. Fatigue and depression improved in most patients after LT regardless of MEQ-SA score, with the exception that the definite morning type tended to do show a worsening of depression. Dyskinesia was worsened in all but 3 patients and this was also independent of chronotype as determined by MEQ-SA scores.

Table 7 illustrates the relationship between MEQ-SA score and performance on the TMTs after LT. Note the tendency to perform better on the ETF (left columns) and the FTK (right columns) in almost all LT patients and this was independent of the MEQ-SA score achieved. The tendency for a few patients to show a worsened performance on these tests was scattered equally across the range of MEQ-SA scores characterizing this patient sub-group.

Table 8 illustrates that a similar number of patients in the sample tested showed improvement or deterioration in bradykinesia (left columns) and rigidity (middle columns) and this was again unrelated to MEQ-SA scores. However, there was a large number of patients showing an improvement or no change in tremor (right columns) and this is twice the number showing a worsening of tremor. Note that most of these patients scored in the mid to definite morning range of the MEQ-SA. The patients that showed worsening of tremor after LT were equally distributed across the spectrum of MEQ-SA scores characterizing the sample.

### 3.9. RSBD, Dreaming, Compliance and Drug Intake

Other minor parameters examined include RSBD, dream intensity, dream vividity and reduction in drug burden. Only 58% of the patients treated with LT were diagnosed with active RSBD at the time they entered the program. Of these patients 33% showed a reduction in RSBD which was better than their pre-program score, while only 6% showed an increased severity of RSBD during LT. The remaining 61% of the patients with RSBD remained unchanged on this measure for the course of the study. The change in dream intensity was rated as moderate to severe in 60% of the patients before commencing LT. During the course of treatment, this was reduced to a mild rating in 60% of patients, while 20% of patients experienced a dream intensity that went from moderate to severe. The remaining 20% of patients showed no change in this parameter. Dream vividity was reported to decrease in 15% of LT treated patients, while 24% reported an increase in this parameter. Of all treated patients, 61% reported no change in this parameter. There was drug reduction in 51 patients during the time that they remained in the program receiving LT. The types of drugs that were successfully reduced or discontinued included DA replacement, soporifics and anti-depressants during the course of the study.

As experienced previously [7,8], non-compliance is an ongoing problem with LT, with the drop-out rate of non-compliant patients occurring earlier in the program and contributing significantly to the high rate of natural attrition characterizing LT. There were many factors related to non-compliance, occurring much more frequently in patients that responded poorly to the treatment than those that responded well. Informal records on compliance factors in this sample of patients suggest that, while 29% of non-responders demonstrated poor compliance, only 8% of responders demonstrate this feature. Non-compliance was a common feature of patients that quit the program early. The yearly attrition of patient numbers from the program is expressed in a Appendix A for each of the 10 years of the study.

## 4. Discussion

### 4.1. Sleep, RSBD and Dreaming

The present results demonstrate improvement in sleep after the application of evening light for a period of up to 10 years in patients with PD. The improvement in sleep included longer sleeping time, fewer awakenings, falling to sleep earlier and consolidation of sleep. When the data were analyzed for a group receiving evening LT, there appeared to be little difference in the time asleep and time awake compared to their performance prior to entering the program. This did not change for the course of treatment over 10 years for the group as a whole. However, when each patient within the group was classified according to when they fell asleep and when they awoke, two distinct groups became evident: those that fell asleep or awoke earlier and those that fell asleep and awoke later than they did prior to commencing LT. This question initially arose because we were curious from whence their gained sleep was derived. Given that PD patients experience advanced sleep phase syndrome [10], evening LT was applied to delay their sleep phase causing later sleep onset. However, while some patients (57–67%) experienced a phase delay and fell sleep later in the evening, 33–43% fell sleep earlier, which is inconsistent with the phase shift hypothesis [27]. This suggests that the observed increase in sleep was achieved through a number of mechanisms governing time to sleep or to awaken, reduction in the number of awakenings and ability to fall back to sleep. Even though these patients were given evening LT to cause a phase delay [18], evening LT advanced their phase further in more than half of the patients on LT. It is possible that this may have been due to a cognitive based decision driven by an event, outside of circadian control [3].

When other sleep related variables, including sleep duration, number of awakenings and ability to fall back to sleep, were examined a long-term pattern of improvement occurred. In the sample of patients treated here, 82 patients increased the total amount of sleep after LT while 26 patients did not. Similarly, 65 of the patients receiving LT decreased the number of awakenings per night while 43 worsened or demonstrated no change. Furthermore, 56 patients that awakened fell asleep more quickly with evening light exposure while 52 fell asleep later. When a frequency distribution was constructed depicting the number of patients increasing their total amount of sleep after LT compared to those that did not, 37 out of 38 patients were sleeping less than 6 h per night before entering the program. All of these patients increased their total amount of sleep per night during their course of LT. Furthermore, non-responders that did not increase their total sleep time, and were getting at least 6 or more hours of sleep prior to starting the program, did not increase their sleep in response to LT. This finding reveals a major advantage of LT in treating sleep in that it appears to regulate sleep but does not dysregulate it. In other words, LT repairs sleep loss in PD patients with insomnia but does not adversely affect those that are getting adequate sleep. This is an advantage over pharmacological intervention in that wearing off, side effects and rebound effects seen with soporific and hypnotic drugs do not occur with LT. While it has been suggested that daytime fatigue is closely linked to sleep [28], the results reported here confirm this relationship in PD. When dividing fatigued patients into responders and non-responders, 80 patients on LT experienced moderate daytime fatigue and showed a significant reduction in this parameter extending for more than 5 years. Non-responding patients showed an increase in fatigue from the 3rd to 4th year and then quit. It is interesting to note further that the fatigue reported before commencing the program was greater in the responsive compared to the non-responsive groups. A similar relationship in responsive versus nonresponsive patients was observed for other variables, including amount of sleep, number of awakenings and tendency to fall back to sleep, with the most severely impaired showing the most robust therapeutic effect.

While LT improved RSBD, intense and vivid dreaming decreased in some patients while it increased in others. Given the importance of melatonin in dreaming and that nightmares are not uncommon after its administration [29,30] and that light antagonizes melatonin [6], we hypothesized that dreaming would serve as an index of melatonin’s potency and that dreaming would lessen during LT. However, with the exception of the response of RSBD to LT [11], dreaming was not consistently affected by LT, suggesting that different mechanisms may be at play in RSBD and dreaming in PD.

### 4.2. TMTs and Assessment of Motor Function

The effects of evening light on the ability of patients to perform TMTs over 10 years were unexpected. Not only did LT improve the performance on both the upper and lower limbs tasks but more severely impaired patients before LT showed the most pronounced improvement. The incremental nature of recovering motor function is similar to that seen with sleep and continued for the duration of the study. The pattern of improvement was more robust in the ETF test compared to the FTK test and was probably due to the high level of comorbid conditions seen in the elderly, including knee and hip problems. The gradual, incremental rate of improvement in both TMTs is similar to that seen in the deterioration of PD and is similar to the rate of change characterizing circadian involvement [27].

With the exception of two studies undertaken to date [7,11], the longest LT outside those undertaken in our clinic is 3 months [10,13]. The present findings suggest that short term studies using LT may not have detected clinical repair because they did not monitor motor function for periods sufficiently long to permit detection of subtle, incremental improvements. With the present study replicating and extending a five year study published previously [7]; See Appendix A), the rate of improvement seen in the present study is consistent with the slow, incremental rate of change defined previously with the rate of change in circadian phase after light exposure [27]. This suggests to us that the slow incremental rate of NSD degeneration may lie within the confines of long-term circadian control [31]. When compared to the rapid therapeutic effect expected with DA replacement, when the degenerative process is slow and incremental [7,27], an open invitation for treatment complications emerges that may explain why dyskinesia develops [4]. Alternatively, when light is administered daily over the long-term, the pace of reparation occurs at a rate which resembles the loss of functional integrity [4]. Implementation of such a protracted course of treatment with LT is consistent with the therapeutic expectation of DA replacement. Future clinical paradigms should take the protracted, incremental nature of the degenerative process into account in developing new treatment strategies. Furthermore, our work also suggests that LT may slow the degenerative process if it is used on a daily basis for protracted periods. Further long-term studies may reveal more about the legitimacy of such a hypothesis.

### 4.3. Dyskinesia and Primary Symptoms

In view of the problems that typically develop with DA replacement, the effect of long-term LT on dyskinesia deserves consideration. For the purpose of our analysis, we divided LT patients into two groups: those that entered the program without dyskinesia and those that entered with it. Those that entered the program with dyskinesia were responsive to treatment while those in the no dyskinesia group experienced little or no dyskinesia until 2.5 years after entering the program. When dyskinetic patients entering the program were compared to those without it, the severity of dyskinesia remained minimal in the latter group up until the fifth year post LT treatment and never reached a level of severity comparable to that seen in patients with prior dyskinesia. From this, it was inferred that the development of dyskinesia may be delayed or suppressed by 2 years with chronic LT. This is an important finding needing further exploration that addresses a major problem that DA replacement presents for PD patients. This was the principal reason for implementing phototherapy in PD, in a study undertaken in 1996, which remains the least cited article in the phototherapy/PD literature [4].

The effects of long-term phototherapy on the three primary symptoms of PD reveals a profile that is similar to that reported previously [7]. While rigidity and bradykinesia improved for the first 3–5 years, they levelled off or increased in severity afterwards for the remaining time. Tremor, on the other hand, continued to improve up until the 6th year after entering the program and then became more variable while levelling off and showing little change after improvement. While about half of the patients showed a remarkable improvement in bradykinesia the other half showed a more progressive deterioration. We suggest that advancing age and not PD per se may also be contributing to this, as the elderly have a tendency to slow down as age advances. Improvement in tremor was seen in 73 patients, while 22 patients continued to deteriorate on this parameter. It is interesting that, as observed with prodromal symptoms, the most responsive patients were those that exhibited the most severe deficits. This phenomenon has been confirmed in early LT studies, one of which was a controlled trial [5,7].

### 4.4. Parkinsonian Depression and Total Drug Burden

The effect of chronic LT on PD depression in the present study is consistent with the role of the circadian system in the modulation of mood and with its comorbidity routinely reported in this disorder [7,32,33]. As observed with the prodromal symptoms of sleep, the anti-depressant effect of light occurs readily within the first month of commencing treatment and then continues to improve as long as LT is maintained. When the responders are compared to those that had no depression prior to commencing the study, the latter group show only minor deterioration for the 10 year term, as long as LT was maintained. As with dyskinesia, depression does not become as severe as that seen in those patients where depression was present when the program started. Depression may well be the most prevalent psychiatric manifestation of the disease but its position in the order of events linking primary and secondary symptoms is one of the most important findings of the present work.

There are several studies showing that PD patients reduced their total drug burden when treated with adjuvant LT [4,6,7,33]. The first method employed included a drug holiday followed by reintroduction of drug by titration accompanied by daily light exposure [4]. The second method implements the reverse titration of drug in patients already undergoing phototherapy [7]. While increased drug intake was seen in many patients in the present study, at least 40 of the subjects in the present study were able to reduce their DA replacement and maintained a good therapeutic response. Another seven patients were able to eliminate or substantially decrease their antidepressants drugs with another six eliminating their intake of soporific drugs. While it was not the purpose of this study to examine light and drug combinations, this is a critical area of clinical pursuit that would benefit most PD patients given the high rate of DA replacement overdosing, adverse side effects and the regular occurrence of polypharmacy.

### 4.5. Circadian Function in Parkinson’s Disease

The results of circadian phase typing of 26 patients in the available sample using the MSQ-SA revealed that the findings were consistent with the hypothesis that circadian function mediates at least some of the recovery seen after long-term LT (Table 3). The first significant finding is that all patients in our selected study population were scored as intermediate or morning phase type. Few intermediate phase chronotypes, but no evening chronotype patients, in the sample of patients used in the chronobiology study is consistent with previous work describing PD patients as being phase advanced [17,18,19]. Further to this, when long-term evening light was applied to these patients they tended to show a phase delay in their sleep and their time to awaken moved about an hour later into the morning by the end of the 10 year term of observation. It is important to note that Parkinsonian symptoms that are routinely described as being modulated by circadian function were most profoundly affected by long-term LT. These included sleep duration, depression and fatigue, with the majority of patients showing improvement in these parameters for the duration of the study (81%, 73% and 92%, respectively). However, we also observed improved motor function in the longer term. This occurred in an incremental manner, which reflects the protracted rate at which sleep related parameters improved. This suggests that the sequelae of sleep and motor improvement may be coupled in time with the other circadian related parameters that also repair incrementally when LT is applied [34]. Further to the proposed involvement of circadian function in PD is the finding that 88% of the patients in our chronotyped sample all experienced a worsening of dyskinesia regardless of chronotype. In the same way that impaired DA function can compromise motor related parameters, so too it can interfere with circadian mediated functions, such as sleep, depression and fatigue [12,32,33]. This supports the proposed mechanism by which DA degeneration advances and suggests that the temporal relationship between sleep and the motor response to LT may function to establish the pattern of events, serving as a reliable index of early diagnosis for PD. In addition, given the suggestion that interruption of circadian phase might foster the development of PD [31], early treatment with LT during the prodromal phase may prevent onset and deterioration into the more advanced stages of this disorder.

There are many opposing forces that DA degeneration and circadian intervention exert in the PD patient. Reduced circadian rhythmicity exerting pressure on the DA system over several decades may play a role in the onset of DA degeneration. This may intimate early intervention using LT to slow the degenerative process at a rate similar to that seen with the LT-induced improvement in TMTs (Figure 4). As the disease advances, bad timing of DA delivery within the L/D cycle, existing comorbid states, polypharmacy and the process of ongoing DA degeneration could all contribute to the progressive nature of degeneration. While some recovery of function can occur, the degree of recovery will be limited by the amount of DA deficiency that existed prior to commencing treatment and this remains problematic, since 80% of the system is absent at the time of diagnosis [35]. While DA replacement is the long-established treatment providing symptomatic improvement, there is much to be said for intervening in circadian function to bolster the therapeutic effects that DA replacement cannot always provide. This includes sleep restoration, repair of RSBD, reduced Restless Leg Syndrome (RLS) and depression. Reducing the dose of DA replacement accomplished with adjuvant LT can decrease adverse side effects and may even help to slow the degenerative process. While melatonin replacement has not been proven effective as a therapeutic in PD [36,37,38], LT-induced intervention into circadian function through its natural route, via the retina, may be the most effective means of CNS delivery. Some studies have observed adverse effects of melatonin in models of PD [39] and in the disease itself (See in [40]), while LT demonstrates superior therapeutic effects over melatonin. We stress that its action as a melatonin antagonist may be the crucial mechanism at work [39,41].

### 4.6. LT and Compliance

The present results suggest that the most important variables to monitor during the course of treatment with LT is compliance. After more than 25 years of experience in implementing LT for PD, assisting patients in conforming to effective treatment regimens and appropriate device use are important issues for optimizing effective treatment [6]. The present findings confirm the importance of compliance in an array of issues, including time and continuity of use, type of device employed and the position of the device during treatment. If compliance is not monitored on a regular basis then patients drift away from effective regimens, become discouraged and return to less effective routines as their condition deteriorates. In this and in other studies, we found that patients terminate their involvement earlier if they cannot perceive improvement in their condition. The incremental improvement seen with circadian intervention, as described earlier, discourages compliance and promotes early departure from the program if the patient does not perceive the improvement achieved. Appendix A demonstrates the rate of patient drop out for each year of the 10 year program examined in the present study. There was a drop-out rate ranging from 17% to 47% of the remaining subjects for each consecutive year with an average drop-out rate of 29% per annum. In many instances this can be a product of anosognosia, which PD patients can develop early in the course of their disease. In that state, they can sense neither clinical improvement nor deterioration in condition. Proper placement of a light emitting device to encourage optimal phototransduction, coherence to the prescribed treatment regimen and regular clinical feedback is important to explore in future studies to minimize patient drop-out. Informal observations to date suggest that, when the degree of compliance is monitored in relation to their therapeutic response, non-compliance may go hand in hand with treatment non-responsiveness. This is why clinician-to-patient feedback with LT is critical for achieving a good clinical outcome. This remains essential when dealing with symptoms that improve in minimal increments over time and is this subject of our ongoing work.

### 4.7. Limitations, Future Directions and Concluding Remarks

There are several areas of interest evolving from the present findings that are at the focus of future research to determine the most effective use of LT in PD. For example further research on compliance with critical light treatment regimens, detailed studies on the role of patient expectation, and effective light/drug combinations all need to be determined to understand how LT can be most effectively applied as an adjuvant treatment.

There are three features of the non-responsive versus responsive patients that, in many cases, makes them identifiable soon after entry into the program. In the first instance, non-responsive patients find it difficult to adhere to the prescribed regimen of light treatment (i.e., daily time of use, duration of exposure, type of light emitting device employed and compatibility with social life) and they often practice self-determined variations in prescribed and over-the-counter drug regimens. Conversely, responsive patients frequently practice the prescribed anti-PD drug regimen and closely adhere to the light treatment paradigm prescribed by the clinician. The second discriminating feature is that responding patients often show therapeutic improvement in prodromal symptoms (i.e., insomnia, RSBD, depression anxiety, etc.) early within the first 4 weeks of treatment while motor symptoms take longer to manifest in the non-responsive patient. Finally, the most responsive patients were those that showed the most severe deficits at the time of admission into the program. While this phenomenon was most pronounced for tremor and the prodromal symptoms, it is possible that lack of perceived therapeutic effect arising from anosognosia contributed, while those that perceived improvement remained in the program longer. This could account for the mixed clinical effects in the present study and is the focus of ongoing research.

With circadian function becoming an increasingly focus of PD research [2], the therapeutic effects seen here and in other studies employing LT [6,7,8,11,12] suggest that more in-depth exploration of circadian function is needed. This would include melatonin secretion in early onset versus late onset PD, circadian actimetry, the effect of LT upon dim light melatonin onset (DLMO), circulating melatonin levels, as well as critical times of LT application, as important areas deserving investigation. At the present time, with LT treatment being the most effective means of modifying circadian function, the opportunity is open for understanding other neuroendocrine features underlying our knowledge of basic principles of PD. Such principles include why there is a predominance of males versus females afflicted with PD or whether male/female differences exist in response to long-term LT treatment. It is also feasible that the effects of light, other than circadian, may be involved in the observed therapeutic effects. Indeed, the recent observation that altered melanopsin function by LT fails to elicit a robust effect on PD [42] or that the tonic rather than phasic effects of light may be involved in the observed therapeutic effects [43] may well be important factors yet to be explored. Understanding the role of circadian and non-circadian functions of LT from these perspectives will give new direction for producing more effective treatments.

There are several weaknesses in the present work that the reader should be aware of. The first is a lack of inclusion and exclusion criteria. We acknowledge the importance of these in increasing the power of the sample size and the probability of detecting a clinically relevant effect. However, we purposely opened the inclusion criteria while limiting exclusions to enable us to make a statement about the generalizability of LT across a broader population of PD patients. Secondly, patients and clinicians were not blinded to the protocol. Nevertheless, the performance on the TMTs was inherently blinded by the procedural limitations preventing patients from accessing numerical values describing their performance, or the unavailability of any other patient scores for use in analysis during the 10 year course of the study. Not implementing formal scales of PD assessment in the present study does produce some problems in making comparisons across other published work. However, the inherent problems associated with clinical assessment for most parameters used in the present study are duly recognized and the assessment procedure for evaluating motor function in the present study was based on the criteria defined in the U.P.D.R.S. The Likert Scale used here has been implemented previously on several occasions and in parallel with other standardized forms of assessment [7,8]. In consideration of the limited value of the U.P.D.R.S. in tracking the progress of progressive degeneration in the longer term [20], we were compelled to develop a practical and easily applied method for undertaking the long-term evaluation and scoring of the primary symptoms. This method includes provision of input from carer and the patient in the context of structured evaluation by the clinician. In consideration of the reasonable level of agreement between these results and those published previously [7] we propose that our method of clinical evaluation is reasonably valid and reliable as is the reproducibility of our findings. The validity of the clinical assessments employed in the present study can be derived from parallel use with formal scales such as the U.P.D.R.S and others in an earlier double blind placebo controlled trial [8]. (See randomizing and blinding section in Methods). Taken in the context of the present study, we expect that many of the findings reported here will be critically examined in further randomized, blinded, placebo controlled trials implementing more formal assessments. In particular, the finding dealing with altered circadian function will be the focus of future work.

This study reveals several novel concepts regarding the involvement of the circadian system in various aspects of PD. There are now several components within the symptom matrix of PD that extend far beyond DA deficiency and the replacement of DA. There is a growing body of evidence that the circadian system is also involved and that methods that stimulate the circadian system can improve the primary and secondary symptoms of the disorder. Indeed, this approach suggests that the deeply ingrained and pervading attitude that a single treatment acting by a singular mechanism of action will cure the disease lacks empirical common sense. Such a position does not take into account the complexity of function that the NSD, the circadian system and other interactive systems within the CNS share. Evidence that the circadian system plays an important part in the loss of motor and non-motor control characterizing PD is mounting rapidly [3,44] and a new approach, involving treatment of these symptoms through means other than DA replacement, is becoming a reality. When the PD symptoms that are potentially treatable using chrono-therapeutics are taken into consideration, the treatment approach for PD broadens remarkably and treatment becomes less invasive. We suggest that the in-depth findings emerging from LT studies give a starting place that will provide biologically compatible, more effective treatment approaches that are easily managed and provide a better quality of life for PD patients. No longer can the treatment of PD be examined solely in the context of DA treatment alone [45]. Involvement of the circadian system now commands a multi-directional treatment approach with an emphasis on repairing circadian function in parallel with DA replacement.

## 5. Conclusions

The present study presents long-term data on how the circadian system is compromised in Parkinson’s disease and how circadian intervention using LT can participate in long-term symptomatic control. It also shows that LT, as an adjuvant treatment, can improve extended prodromal symptoms and reduce side effects attributable to DA replacement. As long as patients are compliant with daily treatment in the longer term, the improvement is incremental, depicting a slow rate of progression similar to that seen with progressive DA degeneration. Indeed, this is the critical observation of the present work demonstrating a rate of degenerative progression that is strikingly similar to the rate of light induced circadian restoration.

## Figures and Tables

**Figure 1 brainsci-14-01218-f001:**
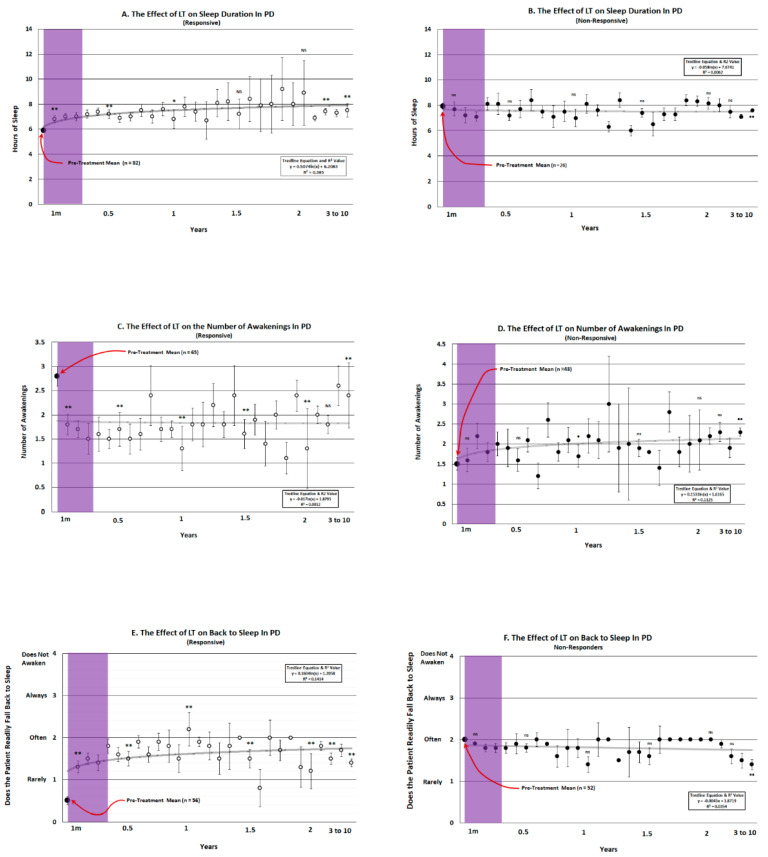
The total number of hours that patients slept (**A**,**B**), the number of awakenings (**C**,**D**) and the tendency to fall back to sleep (**E**,**F**) during the dark phase of the L/D cycle prior to (Large black dot-Left) and after (small dots) undergoing light treatment (LT) for a period of up to 10 years. The pre-treatment mean is expressed for each parameter is marked with a red arrow and the number of patients in each group is indicated in parentheses. Patients were placed into one of two possible categories depicting whether they responded to LT by increasing their average amount of sleep, decreasing the number of awakenings or falling asleep more readily than their performance prior to LT (responsive-open circles). Those patients that decreased their average performance on all three parameters compared to the pre-treatment value were designated as non-responsive (closed circles). Non-parametric testing at representative times during the 10 year period of observation was implemented using the Related Measures Wilcoxon Signed Rank Test (IBM-SPSS^®^-2022) with Bonferroni Correction to analyze differences between pre and post treatment performance. Asterisks indicate a significant improvement in performance (* = 0.05; ** = 0.001) while squares represent a significant deterioration (▪ = 0.05 ▪▪ = 0.001). Values at times tested that were not significantly different to pre-LT treatment were denoted as NS for responsive patients versus ns for non-responsive patients. A logarithmic trend line for the 10 year course of treatment was fitted and is expressed in feint grey for the responders and red for the non-responders with the trend line equation and R2 value denoted in the rectangular box. The translucent purple band represents the approximate time frame that the majority of LT studies have been conducted. The T-bars represent the standard error of the mean and the trend line equation and R2 value are expressed in the rectangular box.

**Figure 2 brainsci-14-01218-f002:**
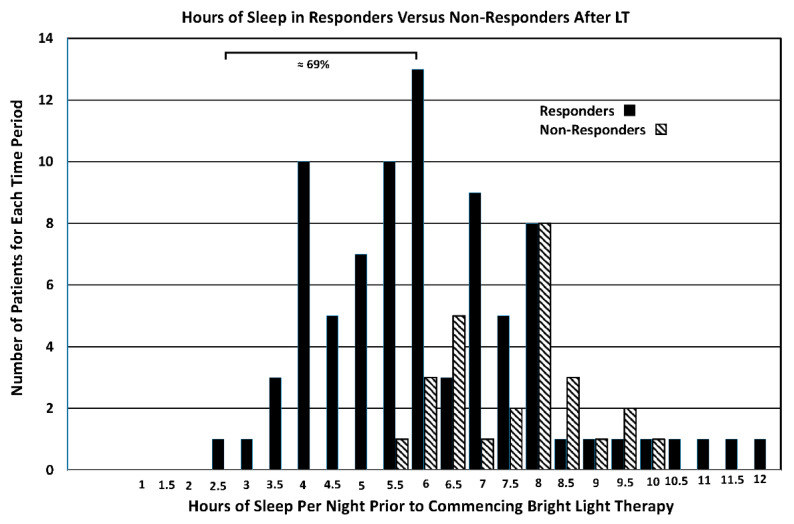
A frequency distribution depicting the number of patients that increased the hours of sleep per night after LT (Responders: closed bars) versus those that did not (Non-Responders: diagonal stripes). The values on the abscissa indicate the number of hours slept by each patient prior to commencing LT. Note that all patients getting 5 h of sleep, or less per night were responders and increased their total amount of sleep per night in response to LT. As seen in prior studies (5 and 7), the most severely impaired patients were the most responsive to treatment. Patients were placed into one of two possible categories depicting whether they responded to LT by increasing their average amount of sleep compared to their performance prior to LT. Those patients that decreased their amount of sleep compared to the pre-treatment value were designated as non-responsive. Note that all patients sleeping 5 h or less each night increased their total amount of sleep after LT treatment.

**Figure 3 brainsci-14-01218-f003:**
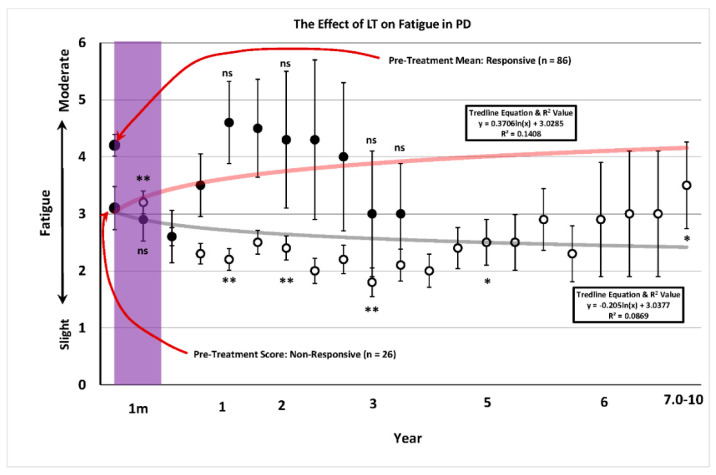
The severity of fatigue prior to (large black dot-left) and after (small dots) undergoing light treatment (LT) for a period of up to 10 years. The pre-treatment mean is expressed with a red arrow and the number of patients in each group is indicated in parentheses. Patients were placed into one of two possible categories. The first consisted of patients that showed a decrease in fatigue compared to the pre-treatment level prior to LT (responsive: open circles). The second group were those patients that showed increased fatigue compared to the pre-treatment value and these were designated as non-responsive (closed circles). Non-parametric testing at representative times during the 10 year period of observation was implemented using the Related Measures Wilcoxon Signed Rank Test (IBM-SPSS^®^-2022) with Bonferroni Correction to analyze differences between pre and post-treatment performance. Asterisks indicate a significant improvement in performance (* = 0.05; ** = 0.001). Values at times tested that were not significantly different to pre-LT treatment were denoted as “ns” for non-responsive patients. A logarithmic trend line for the 10-year course of treatment was fitted and is expressed in feint grey for the responders and red for the non-responders with the trend line equation and R^2^ value denoted in the rectangular box. The translucent purple band represents the approximate time frame in which the majority of LT studies have been conducted. The T-bars represent the standard error of the mean and the trend line equation and R^2^ value for each trend line are expressed in the rectangular box.

**Figure 4 brainsci-14-01218-f004:**
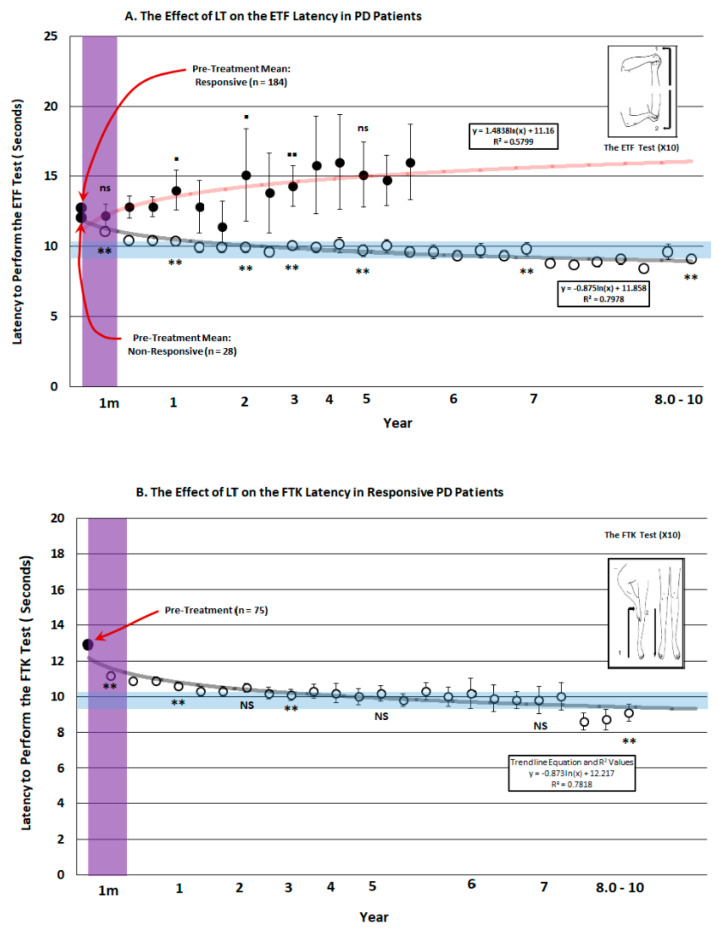
The performance of PD patients on timed motor tests (TMT) including latency to perform the elbow to fist test (ETF: **A**) or the floor to knee test (FTK: **B**,**C**) prior to (large black dot-left) and after (small dots) undergoing light treatment (LT) for a period of up to 10 years. Patients were placed into one of two possible categories depicting whether they responded to treatment by decreasing the time required to perform the TMTs (closed circles) versus those that slowed in performing the TMTs (open circles). The control performance is expressed as the pre-treatment mean prior to LT. Non-parametric testing at representative times during the 10 year period of observation was implemented using the Related Measures Wilcoxon Signed Rank Test (IBM-SPSS^®^-2022) with Bonferroni Correction to analyze differences between pre and post-treatment performance. Asterisks indicate a significant improvement in performance (** = 0.001); Squares indicate a significant decrement in performance (▪ = 0.05 ▪▪ = 0.001). Values at times tested that were not significantly different to pre-LT treatment were denoted as “ns” for non-responsive patients. A logarithmic trend line for the 10 year course of treatment was fitted and is expressed in feint grey for the responders and red for the non-responders with the trend line equation and R^2^ value denoted in the rectangular box. The translucent purple band represents the approximate time frame in which the majority of LT studies have been conducted. The T-bars represent the standard error of the mean and the trend line equation and R^2^ value for each trend line are expressed in the rectangular box.

**Figure 5 brainsci-14-01218-f005:**
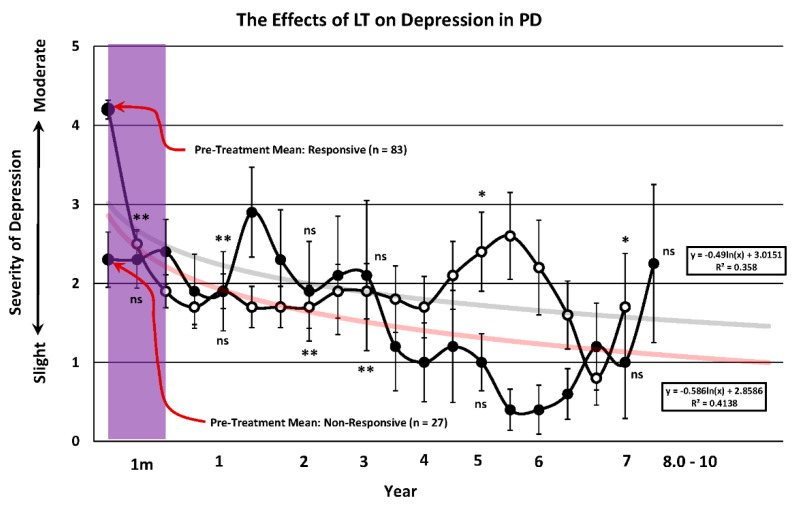
The severity of depression prior to (large black dot-left) and after (small dots) undergoing light treatment (LT) for a period of up to 10 years. The pre-treatment mean is expressed with a red arrow and the number of patients in each group is indicated in parentheses. Patients were placed into one of two possible categories. The first consisted of patients that showed a decrease in depression compared to the pre-treatment level prior to LT (responsive: open circles). The second group were those patients that showed increased depression compared to the pre-treatment value and these were designated as non-responsive (closed circles). Non-parametric testing at representative times during the 10 year period of observation was implemented using the Related Measures Wilcoxon Signed Rank Test (IBM-SPSS^®^-2022) with Bonferroni Correction to analyze differences between pre and post-treatment performance. Asterisks indicate a significant improvement in performance in responsive patients (* = 0.05; ** = 0.001) while ns denotes no significant difference in non-responders compared to pre-LT treatment values. A logarithmic trend line for the 10 year course of treatment was fitted and is expressed in feint grey for the responders and red for the non-responders with the trend line equation and R^2^ value denoted in the rectangular box. The translucent purple band represents the approximate time frame in which the majority of LT studies have been conducted. The T-bars represent the standard error of the mean and the trend line equation and R^2^ value for each trend line are expressed in the rectangular box.

**Figure 6 brainsci-14-01218-f006:**
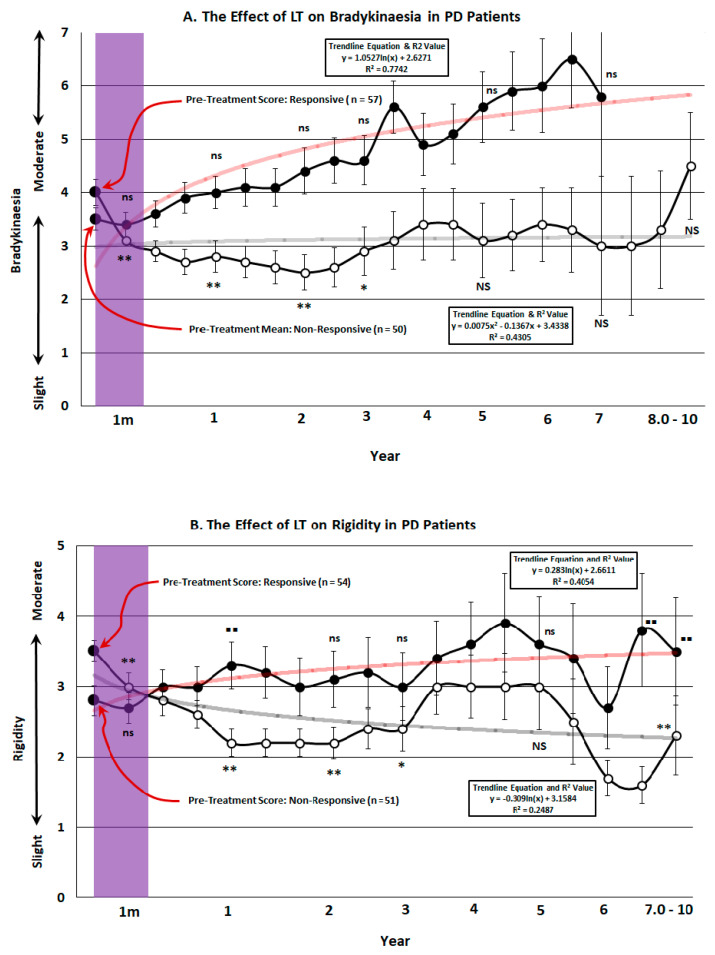
The severity of bradykinesia (**A**), rigidity (**B**) and tremor (**C**) prior to (large black dot-left) and after (small dots) undergoing light treatment (LT) for a period of up to 10 years. The pre-treatment mean is expressed with a red arrow and the number of patients in each group is indicated in parentheses. Patients were placed into one of two possible categories. The first consisted of patients that showed a decrease in any of these three parameters compared to the pre-treatment level prior to LT (responsive: open circles). The second group were those patients that showed an increase in each parameter compared to the pre-treatment value and these were designated as non-responsive (closed circles). Non-parametric testing at representative times during the 10 year period of observation was implemented using the Related Measures Wilcoxon Signed Rank Test (IBM-SPSS^®^-2022) with Bonferroni Correction to analyze differences between pre and post-treatment performance. Asterisks indicate a significant improvement in performance (* = 0.05; ** = 0.001) while squares represent a significant deterioration (▪ = 0.05 ▪▪ = 0.001). Values at times tested that were not significantly different to pre-LT treatment were denoted as “NS” for responsive patients versus “ns” for non-responsive patients. A logarithmic trend line for the 10 year course of treatment was fitted and is expressed in feint grey for the responders and red for the non-responders with the trend line equation and R^2^ value denoted in the rectangular box. The translucent purple band represents the approximate time frame in which the majority of LT studies had been conducted. The T-bars represent the standard error of the mean and the trend line equation and R^2^ value for each trend line are expressed in the rectangular box.

**Figure 7 brainsci-14-01218-f007:**
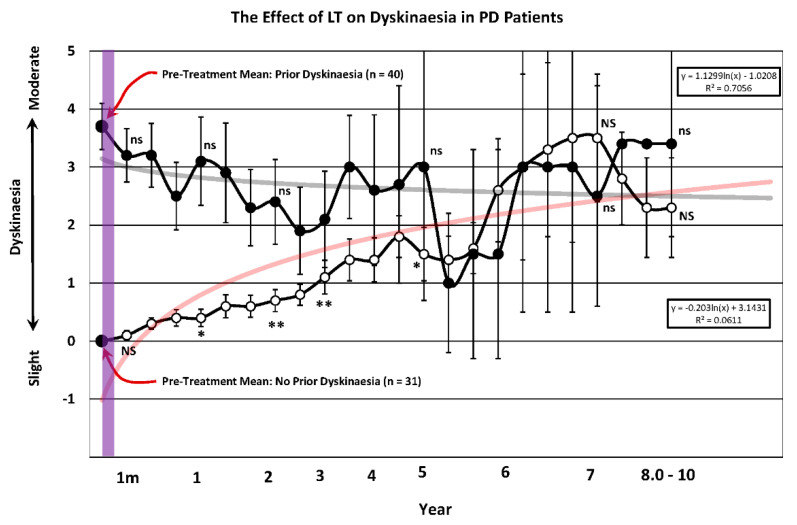
The severity of dyskinesia prior to (large black dot-left) and after (small dots) undergoing light treatment (LT) for a period of up to 10 years. The pre-treatment mean is expressed with a red arrow and the number of patients in each group is indicated in parentheses. Patients were placed into one of two possible categories. The first was patients that showed no dyskinesia prior to entering the program (open circles) while the second group were those patients that showed an increase in dyskinesia after entering the program compared to the pre-treatment value (closed circles). Non-parametric testing at representative times during the 10 year period of observation was implemented using the Related Measures Wilcoxon Signed Rank Test (IBM-SPSS^®^-2022) with Bonferroni Correction to analyze differences between pre and post-treatment performance. Asterisks indicate a significant increase in dyskinesia (* = 0.05; ** = 0.001). Values at times tested that were not significantly different to pre-LT treatment were denoted as “NS” for no prior dyskinesia patients versus “ns” for patients with dyskinesia before entering the program. A logarithmic trend line for the 10 year course of treatment was fitted and is expressed in feint grey for the patients with prior dyskinesia and red for patients no prior dyskinesia. The trend line equation and R^2^ value are denoted in the rectangular box. The translucent purple band represents the approximate time frame in which the majority of LT studies have been conducted. The T-bars represent the standard error of the mean and the trend line equation and R^2^ value for each trend line are expressed in the rectangular box.

**Table 1 brainsci-14-01218-t001:** The demographics of patients enrolled in a 10 year study examining the effects of LT on the symptoms of PD. Age, time remaining in the program, and L-Dopa equivalent dose [16] on start are expressed as the mean ± the standard deviation. The number of male versus female patients are expressed in actual numbers and percent male vs. female. The range in ages of males and females is also expressed in years.

Parameter	Males	Females
Number	75 (63%)	44 (37%)
Age	68.7 ± 9.1 Years	68.1 ± 8.8 Years
Range	49–87 Years	48–83 Years
Time Remained in Program	31.6 ± 26 Months	30.4 ± 29.7 Months
L-Dopa Equivalent Dose on Start	750.7 mg/Day	529.6 mg/Day
MEQ-SA Patient Sample Size	19 (72%)	7 (28%)

**Table 2 brainsci-14-01218-t002:** The time to fall asleep and to awake ascertained from sleep diaries (A,B) or from a circadian chart during completion at clinical interview (C,D) in patients that had undergone LT. Where patients demonstrate advanced phase (falling asleep or awakening earlier) or delayed phase (falling asleep or awakening later), the units are expressed in minutes (min) while the times to sleep and to awaken are expressed in real time ± SD. The percentage of patients being phase delayed or phased advanced for each measure are expressed in the second column from the left. The actual time to sleep or to awaken after LT (right column) was determined by adding the obtained ∆ in time to sleep or to awaken to the respective mean real time to sleep or awaken (second column from the right) prior to LT.

**A. Sleep Diary—Time to Sleep: Over Days 1 to 42**				
		**∆** **in Time**	**Time to Sleep**	**Time to Sleep**
	**% of Group**	**to Sleep**	**Prior to LT**	**After LT**
Advanced Phase	43%	60 m Earlier	11:25 p.m. ± 18 m	10:25 p.m.
Delayed Phase	57%	30 m Later	10:15 p.m. ± 9 m	10:45 p.m.
**B. Sleep Diary—Time to Awaken: Over Days 1 to 42**				
		**∆** **in Time**	**Time to Wake**	**Time to Wake**
	**% of Group**	**to Wake**	**Prior to LT**	**After LT**
Advanced Phase	43%	43 m Earlier	07:12 a.m. ± 14 m	6:29 a.m.
Delayed Phase	57%	54 m Later	6:02 a.m. ± 12 m	7:56 a.m.
**C. Circadian Chart—Time to Sleep: Over 10 Years**				
		**∆** **in Time**	**Time to Sleep**	**Time to Sleep**
	**% of Group**	**to Sleep**	**Prior to LT**	**After LT**
Advanced Phase	46%	30 m Earlier	10:50 p.m. ± 9 m	10:20 p.m.
Delayed Phase	54%	27 m Later	10:00 p.m. ± 10 m	10:27 p.m.
**D. Circadian Chart—Time to Awaken: Over 10 Years**				
		**∆** **in Time**	**Time to Wake**	**Time to Wake**
	**% of Group**	**to Wake**	**Prior to LT**	**After LT**
Advanced Phase	30%	45 m Earlier	7:25 a.m. ± 13 m	6:40 a.m.
Delayed Phase	70%	53 m Later	6:00 a.m. ± 11 m	6:53 a.m.

**Table 3 brainsci-14-01218-t003:** The demographics of dyskinetic patients that underwent LT. The time related descriptions of patients with dyskinesia are exhibited in the left column with the presence of pre-trial versus no pre-trial dyskinetic state expressed in the middle column and the years at each stage expressed in the right hand column. The time to onset of moderate dyskinesia for the group entering the program with pre-existing dyskinesia was calculated from the rate of development exhibited by patients that entered the program without dyskinesia. However, it is important to note that the intensity and rate of development of dyskinesia in those patients is reduced, because they underwent phototherapy as dyskinesia developed. Note that the “nearest” and “estimated” values in the right hand column were determined from pre-entry values.

Age of Patients in at Diagnosis	Pre-Trial Dyskinesia	60.4 ± 10.3 years
	No Pre-Trial Dyskinesia	61.2 ±12.6 years
Age at Start of Program	Pre-Trial Dyskinesia	67.9 ± 8.4 years
	No Pre-Trial Dyskinesia	65.8 ± 9.2 years
Years to Enter Program After Diagnosis	Pre-Trial Dyskinesia	7.6 ± 4.9 years
	No Pre-Trial Dyskinesia	2.8 ± 2.9 years
Time to Dyskinesia Onset After Diagnosis	Pre-Trial Dyskinesia	4 Years (Estimated)
	No Pre-Trial Dyskinesia	5.5 Years
Time to Onset of Moderate Dyskinesia After Entering the Program	Pre-Trial Dyskinesia	0 Years
	No Pre-Trial Dyskinesia	7 Years (Nearest Value)
Time to Onset of Moderate Dyskinesia from Diagnosis	Pre-Trial Dyskinesia	6 Years (Estimated)
	No Pre-Trial Dyskinesia	9.8 Years

**Table 4 brainsci-14-01218-t004:** A table depicting the chronotype of patients classified using the MEQ-SA assessment tool in relation to the advancement or delay of their time to sleep or to awake after a LT duration of up to 10 years of LT. The scores appearing in the furthest left column represent the chronotypes of definite morning (D. Morn) moderate morning (M. Morn) or intermediate (Interm.) appearing in the second column. Times to sleep and to awaken were entered after patients recorded their times in their Sleep Diary for the first 40 days in the program. On that basis, they were designated as being advanced (Advan.) or delayed (delay). Data in the circadian chart columns depicts data collected during clinical interviews for the time each patient remained in the program up to 10 years. On that basis, they were again designated as being advanced or delayed as determined by their tendency to their fall asleep or awaken. The percentages of patients in the 26 patient sample for each category of advanced or delayed after LT are expressed in the bottom row.

		Sleep Diary (40 Days)						Circadian Chart (10 Years)				
		To Sleep			To Awaken			To Asleep			To Awaken	
MEQ-SA	C TYPE	Advan	Delay		Advan	Delay		Advan	Delay		Advan	Delay
**73**	**D. Morn**											
**70**	**D. Morn**											
**70**	**D. Morn**											
**69**	**M. Morn**											
**68**	**M. Morn**											
**67**	**M. Morn**											
**67**	**M. Morn**											
**67**	**M. Morn**											
**66**	**M. Morn**											
**66**	**M. Morn**											
**65**	**M. Morn**											
**64**	**M. Morn**											
**60**	**M. Morn**											
**59**	**M. Morn**											
**58**	**M. Morn**											
**58**	**M. Morn**											
**57**	**M. Morn**											
**55**	**Interm.**											
**54**	**Interm.**											
**54**	**Interm.**											
**54**	**Interm.**											
**51**	**Interm.**											
**51**	**Interm.**											
**50**	**Interm.**											
**50**	**Interm.**											
**49**	**Interm.**											
	**Percent**	**42%**	**58%**		**23%**	**77%**		**46%**	**54%**		**31%**	**69%**

**Table 5 brainsci-14-01218-t005:** A table depicting the chronotype of patients classified using the MEQ-SA assessment tool in relation to the improvement or deterioration in the total amount of sleep per night, the number of awakenings and the ability to fall back to sleep. The MEQ-SA scores are represented in the furthest left column with the chronotypes of definite morning (D. Morn) moderate morn (M. Morn) or intermediate (Interm.) shown in the second column. The data were collected during structured clinical interviews and entered on the circadian chart for a time up to 10 years while the patient remained in the program. The percentages of patients in the 26 patient sample for each category of more or less, better or worse after LT are expressed in the bottom row.

		Amount of Sleep			Awakenings			Back to Sleep	
MEQ-SA	C TYPE	More	less		Fewer	More		Better	Worse
**73**	**D. Morn**								
**70**	**D. Morn**								
**70**	**D. Morn**								
**69**	**M. Morn**								
**68**	**M. Morn**								
**67**	**M. Morn**								
**67**	**M. Morn**								
**67**	**M. Morn**								
**66**	**M. Morn**								
**66**	**M. Morn**								
**65**	**M. Morn**								
**64**	**M. Morn**								
**60**	**M. Morn**								
**59**	**M. Morn**								
**58**	**M. Morn**								
**58**	**M. Morn**								
**57**	**M. Morn**								
**55**	**Interm.**								
**54**	**Interm.**								
**54**	**Interm.**								
**54**	**Interm.**								
**51**	**Interm.**								
**51**	**Interm.**								
**50**	**Interm.**								
**50**	**Interm.**								
**49**	**Interm.**								
	**Percent**	**81%**	**19%**		**58%**	**42%**		**58%**	**42%**

**Table 6 brainsci-14-01218-t006:** A table depicting the chronotype of patients classified using the MEQ-SA assessment tool in relation to the improvement or deterioration in fatigue, depression and dyskinesia. The scores are represented in the furthest left column with the chronotypes of definite morning (D. Morn) moderate morn (M. Morn) or intermediate (Interm.) in the second column. The data were recorded during the structured clinical assessment for as long as the patient remained in the program up to 10 years. The percentages of patients in the 26 patient sample showing improvement (better) or deterioration (worse) for these 3 parameters after LT are expressed in the bottom row.

		Fatigue			Depression			Dyskinaesia	
MEQ-SA	C TYPE	Better	Worse		Better	Worse		Better	Worse
**73**	**D. Morn**								
**70**	**D. Morn**								
**70**	**D. Morn**								
**69**	**M. Morn**								
**68**	**M. Morn**								
**67**	**M. Morn**								
**67**	**M. Morn**								
**67**	**M. Morn**								
**66**	**M. Morn**								
**66**	**M. Morn**								
**65**	**M. Morn**								
**64**	**M. Morn**								
**60**	**M. Morn**								
**59**	**M. Morn**								
**58**	**M. Morn**								
**58**	**M. Morn**								
**57**	**M. Morn**								
**55**	**Interm.**								
**54**	**Interm.**								
**54**	**Interm.**								
**54**	**Interm.**								
**51**	**Interm.**								
**51**	**Interm.**								
**50**	**Interm.**								
**50**	**Interm.**								
**49**	**Interm.**								
	**Percent**	**92%**	**8%**		**73%**	**27%**		**12%**	**88%**

**Table 7 brainsci-14-01218-t007:** A table depicting the chronotype of patients classified using the MEQ-SA assessment tool in relation to performance on timed motor tests (TMT) including the elbow to fist (ETF) test and the floor to knee (FTK) test during 10 years of LT. The MEQ-SA scores are represented in the furthest left column with the chronotypes of definite morning (D. Morn) moderate morn (M. Morn) or intermediate (Interm.) ratings expressed in the second column. The data were collected during the clinical assessment with ten repetitions of the test timed and recorded. They were then rated as better as or worse than the time required to perform the task during the pre-treatment session (i.e., faster or slower). The percentages of patients in each response category are expressed in the bottom boxed for each category. LT = bilateral.

		TMT ETF					TMT FTK			
		Better		Worse			Better		Worse	
MEQ-SA	C TYPE	Left	Right	Left	Right		Left	Right	Left	Right
**73**	**D. Morn**									
**70**	**D. Morn**									
**70**	**D. Morn**									
**69**	**M. Morn**									
**68**	**M. Morn**									
**67**	**M. Morn**									
**67**	**M. Morn**									
**67**	**M. Morn**									
**66**	**M. Morn**									
**66**	**M. Morn**									
**65**	**M. Morn**									
**64**	**M. Morn**									
**60**	**M. Morn**									
**59**	**M. Morn**									
**59**	**M. Morn**									
**58**	**M. Morn**									
**58**	**M. Morn**									
**57**	**Interm.**									
**55**	**Interm.**									
**54**	**Interm.**									
**54**	**Interm.**									
**54**	**Interm.**									
**51**	**Interm.**									
**50**	**Interm.**									
**50**	**Interm.**									
**49**	**Interm.**									
	**Percent**	**84%**		**16%**			**79%**		**21%**	

**Table 8 brainsci-14-01218-t008:** This table depicts the chronotype of patients classified using the MEQ-SA assessment tool in relation to the improvement or deterioration in bradykinesia, rigidity and tremor with 10 years of LT. The scores are represented in the furthest left column with the chronotypes of definite morning (D. Morn) moderate morn (M. Morn) or intermediate (Interm.) in the second column. The data were recorded during the structured clinical assessment for as long as the patient remained in the program up to 10 years. The percentages of patients in the 26 patient sample showing improvement (better) or deterioration (worse) for these 3 parameters after LT are expressed in the bottom row. BILAT = bilateral; L-only = Left side only; R-only = right side only; NC = no change; N/A = not applicable.

		Bradykinaesia			Rigidity			Tremor		
MEQ-SA	C TYPE	Better	Worse		Better	Worse		Better	Worse	N/C
**73**	**D. Morn**							**R-only**	**L-only**	
**70**	**D. Morn**							**L-only**		**R-NC**
**70**	**D. Morn**							**L-only**		**R-NC**
**69**	**M. Morn**							**BILAT**		
**68**	**M. Morn**							**L-only**		**L-NC**
**67**	**M. Morn**							**R-Only**		** L-NC **
**67**	**M. Morn**							**R-Only**		
**67**	**M. Morn**								**BILAT**	
**66**	**M. Morn**							**BILAT**		
**66**	**M. Morn**								**R-only**	
**65**	**M. Morn**									**BILAT-NC**
**64**	**M. Morn**							**BILAT**		
**60**	**M. Morn**							**BILAT**		
**59**	**M. Morn**								**BILAT**	
**58**	**M. Morn**								**L-only**	**R-NC**
**58**	**M. Morn**								**L-only**	**R-NC**
**57**	**M. Morn**									**R-NC**
**55**	**Interm.**							**R-only**		
**54**	**Interm.**							**R-only**		
**54**	**Interm.**							**R-only**		
**54**	**Interm.**							**L-only**		
**51**	**Interm.**									**BILAT-NC**
**51**	**Interm.**							**R-only**		
**50**	**Interm.**								**BILAT**	
**50**	**Interm.**								**R-only**	
**49**	**Interm.**								**L-only**	
	**Percent**	**46%**	**54%**		**58%**	**42%**		**58%**	**42%**	**N/A**

## Data Availability

The data presented in this study are available on request from the corresponding author it contains information that could compromise the privacy of research participants.

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
