# Peer review of "Circadian Intervention Improves Parkinson’s Disease and May Slow Disease Progression: A Ten Year Retrospective Study"

_brainsci, 2024, doi:10.3390/brainsci14121218_

Round 1

Reviewer 1 Report

Comments and Suggestions for Authors

In this paper, the authors have comprehensively and meticulously presented a decade of data on how the circadian system is impaired in Parkinson's disease and how circadian intervention with LT can provide long-term symptom control. They have adequately demonstrated that LT, as an adjunct treatment, can improve prolonged prodromal symptoms and reduce side effects attributed to DA replacement. The article further emphasizes the impact of patient engagement, awareness of the disease, and adherence to procedures on the long-term effectiveness of treatment. The authors also list and address the weaknesses of the study.

Congratulations on a very interesting article. In my opinion, the research was presented comprehensively. The methods and results were clearly presented. The conclusions are appropriate. 

Minor editorial correction:

table 1and 2A- the "words" are not on the same line

in line no. 109 ":" is missing, between 18:00 and 22:00 alike in line 157

Author Response

In response to Reviewer 1, the values in Table 1 were not aligned. These have now been aligned and are highlighted in green in table 1 on page 3. Similarly, in response to the second comment the values in Table 2 were also aligned and have been highlighted on page 7.
The third comment by this reviewer is that a colon be placed in the times designated on page 3. This has now been corrected and is also highlighted in green.

Reviewer 2 Report

Comments and Suggestions for Authors

The manuscript addresses an interesting and little-studied topic in the field of neuroscience, the use of long-term light treatment to study circadian function in the treatment of Parkinson's disease. Overall, the manuscript is well-structured and written. The combination of clinical relevance and an established approach makes the work a valuable contribution to its field. The methodology used by the authors to develop the work is appropriate. Regarding the cited literature, fundamental studies and reviews that are widely recognized in the study area are mentioned. Including recent references indicates an effort to present an up-to-date manuscript. Minor revisions are suggested:

In the keywords, why is the word fatigue written with an initial lowercase letter?

In graph number 3, on the X axis, it should say 7-10.

The captions of figures 1 and 3 end with a double dot.

In figures 4 and 5, on the X axis, it should say 8-10.

Throughout the text, there are several double spaces between words.

The reference list does not have the format requested by the journal.

Author Response

The first comment by this reviewer is that the word “fatigue” in the key words needs an upper case “F”. This has now been corrected and is highlighted in the Key Words Section on page 1.
In the second comment this reviewer has pointed out that the end of the legends for figures 1, 3 and 4 have 2 full stops. This too has now been corrected and the extra full stop has been removed from the text on pages 10, 11 and 12, respectively. This reviewer has also stated that the label on the X-Axis on Figure 3 should read 7-10 but since this is what the figure already depicts I am not sure to what this reviewer is referring. Similarly this reviewer stated that the label on the X-Axis for figures 4 and 5 should read “8-10” but again the X-Axis in those figures already reads “8-10” this so I am not sure to what the reviewer is referring. However, the explanation of the binning process was not made clear in the methods section (page 6; 1st paragraph) and this has now been modified to add clarity and is highlighted in green.
This reviewer also comments on the number of double spaces appearing throughout the text. However these appear only in the version type set by MDPI and appears to be a product of the left and right justified margin format style used by MDPI. While some can be modified, many are not, in fact, double spaces and cannot be removed. Nevertheless, the MS was vetted to correct the spacing where possible.
Similarly the presentation of references in the MDPUI version of the references was a product of the in-house type setting and these have now been corrected in the reference section. Also the use of parentheses to half round versus square and vice versa has been done throughout the references and text to conform to journal format.

Reviewer 3 Report

Comments and Suggestions for Authors

Dear authors, this study is interesting, and it deals with light interventions on sleep disturbances in individuals with Parkinson's disease with an excellent time window of study, although these aspects are interesting and certainly in innovative it is important to point out that this work needs some modifications before it is considered again for publication, below are my suggestions:

1) the abstract is too verbose, especially in the introductory parts, introduce more numbers and more specific directions.

2) For the introduction you should also consider and cite recent other interesting work on the use of supplements and innovative monitoring for sleep study: https://doi.org/10.1016/j.clineuro.2024.108404

3) it would be interesting to introduce flow-short illustration of the study design, as the materials and methods are also somewhat verbose

4) these patients could be monitored in a highly sensitive and specific manner using PET, in this regard I suggest you read and cite: https://doi.org/10.1016/j.clineuro.2024.108404

That may be enough, I look forward to reading the revised manuscript

Author Response

In line with the first comment by Reviewer 3 the abstract has now been rewritten. The final product is much more succinct and this has improved the presentation of the paper and appears highlighted on page 1.
To accommodate the reviewer on the second point we have now modified the introduction by making reference to other innovated methods used for monitoring sleep in papers cited in the collection of PD/light treatment literature already included in the paper. We thought it would be more appropriate to cite papers relevant to the present work rather that introduce additional references at his point. Nevertheless, this was a good point raised by this referee that has added clarity to the presentation. This is highlighted in green on page 2, second paragraph.
In response to the 3rd point raised by this reviewer, our study design was simple with a 2 to 3 day observation period on baseline performance followed by 10 years of treatment and observations at strategic times. We would kindly suggest that introducing a table for this would do little to add to the clarity of design or its presentation. This is particularly true as the simplistic design of the study is clearly presented in figure 1 and 3-7. We suggest that the addition of yet another table would add unnecessary bulk to the paper.
Finally, while we agree with this reviewer that, as stated in point 4, that the use of PET scan to examine DA would be interesting, this approach is not really on track with the purpose of the paper. Given that the complex logistics and cost of such an undertaking would elevate the task far beyond the capabilities of most research facilities or health systems, we intimate that this suggestion is not tenable in the context of the present work. With the somewhat remote relevance of the suggestion to the aims of the present study we would kindly request that this is not doable.